# Assessing the Mental Health, Physical Activity Levels, and Resilience of Today’s Junior College Students in Self-Financing Institutions

**DOI:** 10.3390/ijerph16173210

**Published:** 2019-09-03

**Authors:** Susan Ka Yee Chow, Edward Kwok Yiu Choi

**Affiliations:** 1School of Nursing, Tung Wah College, Hong Kong, China; 2Teaching and Learning Centre, Lingnan University, Hong Kong, China

**Keywords:** junior college student, resilience, positive mental health, physical activity

## Abstract

In recent decades, the number of adolescents and young adults with poor mental health has been increasing, particularly among students in tertiary institutions. This study investigates the physical activities, resilience, and mental health status of junior college students in Hong Kong. The questionnaire consisted of demographic characteristics, the Positive Mental Health Scale, the Brief Resilience Scale, and the Godin-Shephard Leisure-Time Physical Activity Questionnaire. Four hundred and sixteen students participated in the study. The results showed a moderate positive correlation (*r* = 0.485) between resilience and mental health, and a low positive correlation (*r* = 0.258) between resilience and physical activity. The one-way analysis of variance (ANOVA) with a post hoc test showed that arts students engaged in more physical activity than students from other disciplines. A multiple regression analysis was used to examine the predictors of a positive mental health status. The significant predictors are: resilience (β = 0.704; 95% CI = 0.575–0.833; *P* < 0.001), physical activity score (β = 0.032; 95% CI = 0.016–0.048; *P* < 0.001), the male gender (β = 1.035, 95% CI = 0.171–1.900; *P* < 0.05), and students’ enrollment in a health science discipline (β = 1.052, 95% CI = 0.175–1.930; *P* < 0.05). Preventive measures, such as strengthening resilience, a broad curriculum and taking note of the demographic and cognitive characteristics of students are essential for improving the mental health of freshmen in colleges.

## 1. Introduction

Mental health is operationalized as a group of symptoms illustrating the subjective well-being of an individual. It is about positive functioning in life, and the interaction of social and biological factors in the construction of health and disease [1,2]. Although an individual might have good mental health, he or she may feel sad, unhappy, or unwell, as this is part of a fully lived life. Mental health is therefore conceptualized as a positive affect, as well as a feeling of pleasure and of having a sense of mastery over the environment [3]. The World Health Organization (WHO) defines mental health as ‘a state of well-being in which the individual realizes his or her own abilities, can cope with the normal stresses of life, can work productively and fruitfully, and is able to make a contribution to his or her community’ [4].

In Hong Kong, according to statistics from the Psychiatry Department of the Hospital Authority, the number of young people seeking treatment for mental health problems at public facilities has increased dramatically from 12,500 to 22,300 between 2010 and 2015. The Hospital Authority is a statutory body that manages all of the services at Hong Kong’s public hospitals. A recent review showed that a poorer mental health status among Chinese children and adolescents leads to more favourable attitudes toward suicide, which in turn promotes greater suicidal ideation. It has been estimated that the attitude toward suicide has a significant indirect effect on the relationship between the total mental health status and suicidal ideation [5]. From 2006 to 2016, although the suicide rate increased only slightly among the adolescents and young people aged between 15 and 24 years, the number of teens who reported being depressed was a concern. One of the factors behind such depression could be that our society is not adequately preparing young people to be resilient in the face of failure or rejection [6]. Due to poor personal resilience, young people are unable to deal with mental health problems. On the other hand, physical exercise can alleviate emotional stress and protect individuals against diseases related to chronic stress [7].

While commencing a new chapter in life, such as entering a university or other tertiary institution, freshmen and junior students are at risk of feeling stressed. Even students who performed excellently in high school might lack the confidence to succeed academically in this new stage of their life [8]. Other than academic stress, financial constraints, new relationships, and family pressure, which can cause adolescents to strive hard in a highly competitive environment, are major sources of emotional problems [9]. 

Resilience is defined as a type of self-healing and an ability to bounce back, such that being able to overcome obstacles may result in positive outcomes, including recovering quickly from difficulties and self-recovery. It can also be defined as a bridging concept between coping and development, and promoting accommodative competencies [10,11,12]. Empirical studies have found resilience to be strongly associated with positive mental health, to the extent that it may improve or even enhance the mental health status of college students, other young people, and even nurses [13,14,15]. In the fields of transformational psychology and family psychology, it has been reported that resilience is one of the best predictors of the mental health status of an individual [16]. In summary, people who have a poor stress resilience may be at an increased risk of suffering from a mental illness or from suicidal ideation [16,17]. 

Engaging in regular physical activity is not only of benefit to an individual’s physical functioning and level of fitness, but also has favourable psychological benefits. The Hong Kong Government relays the message from the World Health Organization that adults are recommended to engage in at least 150 minutes a week of moderate-intensity aerobic exercise to promote health and relieve stress [18]. A few studies have found that physical activity has no significant direct relationship or only a minimal effect on depressive symptoms in adolescents, student nurses, and employees [19,20,21]. However, other integrative reviews have reported contrasting findings, where the mental health of clinical populations could be enhanced through physical activity, and physical exercise could ameliorate symptoms of anxiety and depression [22]. On the other hand, physical exercise not only being able to reduce stress, the benefits include improving self-esteem and cognitive functioning, which are crucial elements of positive mental health [23]. A current systematic review showed that physical activity intervention or laboratory-based exercise intervention were found able to improve self-esteem in a majority of the research studies [24]. Regarding exercise and well-being, a review illustrated that exercise, physical activity or physical activity interventions are having positive associations with a better quality of life. Meanwhile, a good quality of life means that people who are able to enjoy life and cope with life events experience energy and vitality, and have social support [25].

Some previous evidence supported the view that physical activity is associated with resilience in different populations, including highly anxious individuals and healthy adults [7,26]. Among undergraduates with high anxiety, physical activity was positively and significantly related to self-perceived resilience, but not among those with moderate or low levels of anxiety [26]. With regard to Hong Kong Chinese adolescents, their physical activity levels were significantly correlated with mental well-being, with resilience being the only significant mediator, contributing to about 60% of the relationship [27]. Having regular exercise not only achieves physical fitness, through the biological mechanism; spontaneous physical activity is able to confer resilience and well-being through improvement in both psychological and physical health [28].

In a competitive learning environment, junior students who have just entered another stage of their life are striving hard to achieve success. Based on the diversity of evidence relating to physical activities, resilience, and mental health, the findings on the relationships among these three variables are inconclusive. While researchers have attempted to explain the relationships between mental health, resilience, physical activities, and academic persistence, the social assumption is that the above variables are indeed connected. There is a paucity of studies on the three dimensions of physical activities, resilience, and mental health in relation to college students and on whether demographic factors contribute to an individual’s mental health status. In Hong Kong and worldwide, self-financing post-secondary tertiary institutions provide flexible and alternative pathways to publicly funded universities for secondary school graduates to continue on to higher education. The background and demographic characteristics of these students could affect their behaviours, life activities, and accomplishments during college life. Research focusing on students in self-financing tertiary institutions could delineate the specific characteristics of these students and address the need to improve the quality of education and learning for this specific group of young people.

Hence, the objectives of the study were to determine: (i) the physical activities, mental health, and resilience of freshmen college students; (ii) the correlations between physical activities, mental health, and resilience among the students; (iii) whether any differences existed between students of various disciplines in terms of physical activities, resilience, and mental health; and (iv) whether demographic factors, physical activities, and resilience are predictors of the mental health status. The results will expand knowledge about the interrelationships among the variables and highlight the predictors of mental health.

## 2. Materials and Methods

### 2.1. Design and Sampling

A cross-sectional study employing a quantitative method of collecting data was designed. The sampling method was convenience sampling, while data were collected using a survey questionnaire. 

This study analysed data from full-time first year bachelor’s degree students studying in self-financing institutions in Hong Kong. In the academic year 2017/2018, the total number of students studying in full-time bachelor’s degree programmes in sixteen self-financing institutions was 13,532 [29]. Students from the three largest institutions were invited to participate in this study, as these three institutions offer a wider range of programmes than the other self-financing colleges. The criteria for inclusion were local students enrolled as freshmen in a full-time bachelor’s degree programme. Sub-degree, part-time, or non-local students, or those who had already completed a bachelor’s degree, were excluded. 

According to Rea (2014), the formula for calculating the sample size for a population is as given below [30]:(1)$$\frac{Z_{a}^{2}\left(0.25\right)\left(N\right)}{Z_{a}^{2}\left(0.25\right)+\left(N-1\right)ME_{p}^{2}}$$
(2)$$\frac{1.96_{}^{2}\left(0.25\right)\left(13,532\right)}{1.96_{}^{2}\left(0.25\right)+\left(13,532-1\right)(0.05_{}^{2})}$$

In the formula, *N* denotes the size of the population, *Z* is a 95% level of confidence, while M indicates the margin of error. In this study, the confidence level was set at 95%, while the margin of error was 5%. With a total population size of 13,532, the estimated sample size was no less than 374. Taking into account a 10% dropout rate, the actual sample size would be 416. 

The collecting of data was carried out in January 2019. Hard and soft copies of the questionnaires were distributed either in classes or through the personal social networks of the researchers, respectively. The study protocol and questionnaires were approved by the Research Ethics Committee of the participating institutions.

### 2.2. Instruments and Procedure

The questionnaire on demographic characteristics was developed by the researchers. The Brief Resilience Scale (BRS), the Godin-Shephard leisure-time physical activity questionnaire (GSLTPAQ), and the Positive Mental Health Scale (PMH-Scale) were used to collect data.

The BRS was used to assess the ability of a person to bounce back or recover from stress or adverse events [12]. The scale consisted of six items in a 5-point Likert scale ranging from strongly disagree to strongly agree. Three of the items were negatively worded and had to be coded in reverse. The scores ranged from 5–30. The total score was obtained by summing up all of the item scores, with a higher score indicating a higher level of resilience. With regard to validity, the BRS score was negatively related to perceived stress and positively correlated with the resilience measures, optimism, and purpose in life. The test-retest reliability in three sample groups ranged from 0.62 to 0.69. The internal consistency was considered good, with the Cronbach’s alpha in four sample groups ranging from 0.80 to 0.91. 

The Chinese version of the BRS was used in this study [31]. The scale was translated into Chinese and administered to more than eight hundred undergraduates from Hong Kong and mainland China. The Cronbach’s alpha values of the two samples were 0.76 and 0.72, respectively. A principal component analysis was conducted for the factor structure. The one-factor structure was supported using a scree plot inspection and eigenvalues. The Cronbach’s alpha in this study was 0.71.

The Godin-Shephard leisure-time physical activity questionnaire (GSLTPAQ) is a simple tool that was designed for conducting self-assessments of leisure-time physical activity [32]. The instrument has been translated into different languages. It is a self-reported assessment by the participants of the frequency with which they engaged in strenuous, moderate, and mild activities for more than 15 min during a 7-day period. The activity score is expressed in units. The following formula was used to calculate the total weekly leisure activity score: Weekly leisure-time activity score = (9 X Strenuous) + (5 X Moderate) + (3 X Mild)(3)

There is a direct relationship between the volume of physical activity and the health benefits derived from engagement in physical activities. An individual scoring 24 units or more is considered physically active, resulting in substantial health benefits. The index was also used to identify adults into categories of active and insufficiently active. The test-retest reliability was conducted with Cohen’s kappa, and the k coefficient was 0.40 (95%CI = 0.21–0.60). The validity of the questionnaire has been confirmed based on the positive associations between the leisure activity score index and physical fitness indicators such as VO2max and energy expenditure scores [33]. In order to collect more accurate data for leisure-time activity, the participants were required to provide their activities record in the past four weeks.

Due to cultural differences, some examples of exercises in the GSLTPAQ were deemed inappropriate for the local population. Those exercises were changed according to the guidelines developed by the Department of Health, Hong Kong [34]. For example, cross-country skiing was changed to jumping rope for strenuous exercise, alpine skiing was changed to tennis for moderate exercise, and snowmobiling was replaced with walking for mild exercise.

The PMH-Scale is a scale designed to measure generally positive mental health. It is a 4-point Likert scale, ranging from 1 (not true) to 4 (true). The scale consists of nine items, each of which is a statement, with the participant being asked to indicate the extent to which the statement reflects his/her own situation. Sample items are: ‘I am often carefree and in good spirits’ and ‘I feel that I am actually well equipped to deal with life and its difficulties’. The possible scores range from 9 to 36. The total score is obtained by summing up the scores for the individual items, with a higher score indicating a higher level of satisfaction with life. The PMH-Scale is a brief instrument and in one single dimension. The scale demonstrated a good convergent and discriminant validity. A Pearson’s Correlation coefficient was used to determine the test-retest reliability of the instrument, with r = 0.81.
The internal consistency of the scale was estimated using Cronbach’s alpha, with the results being >0.84 in different sample groups [35]. The Cronbach’s alpha in this study was 0.92.

To ensure that the questionnaires were culturally relevant to the local participants, the PMH-Scale was examined for content validity and test-retest reliability. Three experts were invited to evaluate the representativeness and relevance of the questionnaire to the local population. The instrument was found to have a content validity score of 0.96, showing evidence of content validity [36]. The test-retest reliability of the instrument was examined by inviting 15 volunteers to complete the same questionnaire two weeks apart. The Intraclass Correlation Coefficient was 0.96, which demonstrated that the scale had a good stability [37].

The process of collecting data was conducted in early 2019. A total of 450 questionnaires were distributed during classes or electronically via emails to the students of three tertiary institutions. The questionnaire was discarded if more than 5% of the questions were not answered. This left 416 valid questionnaires to be analysed, for a response rate of 92.44%. 

### 2.3. Data Analysis

The SPSS Statistics for Windows, Version 23.0 (IBM Corp, Armonk, NY), was used to analyse the data. The demographic data were examined by a descriptive analysis, using the mean and standard deviation. A Pearson’s correlation test was used to evaluate the correlations between the physical activities, resilience, and mental health of the participants. The differences in study variables among students in various disciplines were analysed through a one-way ANOVA with a post-hoc test. A multiple linear regression was used to examine the predictors of the positive mental health status for the participants. The normal probability plot was used to examine the data normality of the dependent variable to meet the assumption before the analysis. The significance was set at 0.05, two tailed test.

## 3. Results

### 3.1. Demographic Characteristics

Of the 416 participants, there were more female students (59.9%) than male. The disciplines that they were studying were categorized into the health sciences, arts, sciences, and social sciences (including business studies). More than half of the participants (55.3%) studied health sciences, and only 11.3% studied an arts discipline. A majority (68.3%) of the students held a part-time job; among these, 27.6% worked 8–16 h per week. In terms of friends, 95.9% of participants reported having 1–6 intimate friends. See Table 1 for details.

### 3.2. Resilience, Mental Health and Physical Activities of Students

Descriptive statistics was used to calculate the mean and standard deviation of the variables. The mean score of the BRS was 18.81/30 (SD 3.33), that of the PMH-Scale was 26.26 (SD 5.12), and that of the GSLTEQ was 44.05 (SD 26.72). See Table 2 for details.

### 3.3. Correlations Between Physical Activities, Mental Health and Resilience

The relationships among the physical activities, mental health, and resilience were examined using Pearson’s correlation coefficient. The results showed a significant positive moderate correlation between resilience and mental health (*r* = 0.485, *P* < 0.01), and a significant positive weak correlation between physical activities and mental health (*r* = 0.258, *P* < 0.01). There was no correlation between physical activities and resilience. See Table 3 for details.

### 3.4. Differences in Physical Activities, Mental Health and Resilience among Students in Different Disciplines of Study

To determine whether there were significant differences in the physical activities, positive mental health status, and resilience among the four groups of students enrolled in different disciplines, a one-way ANOVA with a post-hoc test was used in the analysis. The analysis of variance showed significant differences in physical activity among students in different disciplines (F = 3.838, *P* = 0.01). The difference in scores for mental health approached significance (F = 2.488, *P* = 0.06), while there were no significant differences in resilience (F = 1.389, *P* = 0.25) among the students of the four disciplines. See Table 4 for details.

Post hoc analyses using Bonferroni adjusted alpha levels of 0.0125 per test (0.05/4) to determine whether there were any significant differences in the physical activity between students in different disciplines of study would decrease the likelihood of making a type I error [37]. The results indicated that arts students were engaging in more physical activities than students of other disciplines. See Table 5 for details.

### 3.5. Regression Models for the Predictors of Positive Mental Health Status

Based on the results of the correlation analysis in Table 3, in order to identify the predictors of the positive mental health status in students, a multiple regression analysis was performed to examine the contribution of selected variables to the mental health status. The total score of the PMH-Scale was used as the dependent variable, while the predictive variables were: the total score of the BRS, total score of GSLTPAQ, gender, study disciplines (the disciplines were collapsed into health science and non-health science disciplines, and were coded as 1 and 0, respectively), hours of part-time job per week, and number of intimate friends. The normal probability plot showed that the dependent variable is roughly normally distributed overall. The BRS score, GSLTPAQ, being male, and being a health science student were found to be significant predictors of a positive mental health status, with the adjusted R2 = 0.30. The BRS (β = 0.704; 95% CI = 0.575–0.833; *P* < 0.001) and GSLTPAQ scores (β = 0.032; 95% CI = 0.016–0.048; *P* < 0.001) were positively correlated to the PMH. Being a male student (β = 1.035, 95% CI = 0.171–1.900; *P* < 0.05) and studying a health science discipline (β = 1.052, 95% CI = 0.175–1.930; *P* < 0.05) were associated with having a better and more positive mental health status. All of the significant predictors in the model had *P* < 0.05. A model of the regression analysis is shown in Table 6.

## 4. Discussion

The aim of this study was to examine the mental health, physical activities and resilience of college students in Hong Kong. It has been well documented that mental health in both nonclinical and clinical populations is contributing to the overall and subjective well-being of individuals. A robust base of research evidence will lead to a growing emphasis on mental health promotions among students aimed at removing the barriers that affect their overall mental health status. A key strength of this study was the relatively large sample size that took into consideration demographic and related factors that affect the mental health status of students.

### The Correlations Between Physical Activities, Mental Health and Resilience

This study found a moderate positive correlation between resilience and mental health status. Our findings are corroborated by those of Sagone, who found a positive association between positive attitudes and better psychological health under stressful conditions in a study involving university students [38]. Another study revealed that resilience scores significantly predicted depression scores among a group of women college students in India [39]. 

Regarding the correlations between physical activities and mental health, our results showed a weak correlation between the two variables. The results are comparable to a literature review conducted in the past decade on physical activity and mental health in children and adolescents. Despite evidence of an association between physical activity and mental health in young people, there is a paucity of quality research, and the designs of existing studies are often weak, resulting in only small to moderate effects. Nevertheless, primary studies have demonstrated a consistent association between poor mental health and sedentary screen-time [40]. Notwithstanding the evidence, a current systematic review led to the exceptional finding that the experience of stress could impair an individual’s efforts to become physically active. Psychological stress is a predictor of less engagement in physical exercise and, thus, of a more sedentary lifestyle [41]. Whether the level of stress is affecting the level of activity or whether a low level of activity induces more stress is a question that will require continuous research among different population groups to uncover the mechanisms affecting the two variables. In our study, the Positive Mental Health Scale was used to focus on the broader perspectives of mental health, such as whether the participants enjoyed their life and were satisfied with it, and had a feeling of hope and a capacity to cope with adverse life events. Therefore, whether participation in a physical activity might have a direct effect on improving their overall mental health is open to question. Kovess-Masfety et al. illustrated that mental health is clearly influenced by political and socioeconomic factors and norms [42], and that positive mental health is based on social well-being. Social well-being is measured by social acceptance, social integration, and social contribution [43]. Although several studies have pointed to the beneficial effects of exercise and the reduction of stress on the overall mental health status of the participants, the results were more prominent in clinical studies. It may not be possible to achieve such high correlations among the general population. 

Our results showed no association between resilience and physical activities. Physical fitness is a pathway to resilience and serves as a buffer against stress [44]. The contrasting findings in our study could be due to the large standard deviation in the GSLTPAQ scores, which could have reduced the reliability of the results. The physical activity scores ranged from 0–200, the mean = 44.05, and the standard deviation = 26.72. In Hong Kong, due to the crowded living environment and hectic lifestyles, most city dwellers are unable to engage in regular physical activity. According to the Major Sports Event Committee, a Committee tasked to advise the Government on the resource allocation and hosting policy for major sports events in Hong Kong, the most popular sports in Hong Kong are football, volleyball, dragon boat racing, and table tennis [45]. The above sports activities are team sports, and cannot be performed on an individual basis. For high school students, engaging in team sports should not present a problem, given the compulsory physical education lessons and social networks developed in schools. Freshmen studying in tertiary institutions, however, may lack the social networks or team structures to be able to engage in similar kinds of sports activities. As they progress beyond their freshmen year, however, young people tend to form more connections with their peers and could then alter or increase their levels of exercise.

Regarding the statistically significant differences in physical activity levels among students of different disciplines, it was found that arts students engaged in more physical activities than students of other disciplines. Although our results were consistent with those a US study, which found that students attending a liberal arts college were more physically active than their research university counterparts, the difference could be due to the effect of gender, since in that study there were more male students in the liberal arts college than in the research university [46]. In Hong Kong, health science and engineering students are required to complete more study credits, plus an internship or practicum component to fulfil the requirements of the various professional bodies. Arts students, however, do not need to complete as many credits and are not required to complete a practicum as a condition for graduation. This could partly explain why arts students have a less busy school life, giving them more leisure time to engage in physical activities. A study that directly compared motivations to participate in sports and exercise revealed that positive health, affiliation, challenge, competition, social recognition, and enjoyment are motivations to participate in sports [47]. This may lead to the conclusion that besides the availability of leisure time, other intrinsic and extrinsic factors could affect an individual’s engagement in physical activity. To encourage college students to engage in physical activity, both time and motivational factors should be taken into account when designing physical activities and sports programmes aimed at increasing the prevalence of exercise.

With the current study design, it is not possible to make strong claims about the predictors. Based on our findings, a high BRS score, a high GSLTPAQ score, the male gender, and a health science study discipline are possible predictors of positive mental health. From a previous finding, women behave similarly to men when encountering negative life events. When women are without social support when exposed to life events, they are more vulnerable than men without support [48]. The above finding correlates with our finding that mental health clearly varies across certain demographic and social factors. Starting from the early teenage years, more girls than boys turn to friends for informal support or for help for emotional concerns [49], whereas more boys than girls prefer to seek help from a family member or from professional sources [50]. Parental support was considered a robust and unique predictor of adjustment for both boys and girls [51]. During the transition from high school to studying in a tertiary institution, students do seek support for various issues, including emotional, informational, appraisal, and instrumental, from the different people in their lives [52]. As girls tend to turn to friends for support, their peers may not have the ability to provide intervention strategies that are as effective as those that teachers and parents are able to implement with their students and children. Young people may receive divergent messages about how to resolve their problems, which could worsen the existing situation. For informational support, teachers can modify or adjust their teaching to share knowledge and expertise with the students, resulting in an improvement in the students’ emotional status. Due to differences in the help-seeking behaviour between male and female students, it is important to take note of the demographic and cognitive characteristics of students when advising them to seek help. In the case of major life events and emotional crises, all students should be encouraged to deal with their problems using formal and professional networks, and not simply share their thoughts and perceptions with their friends.

Students studying in health science disciplines have a better mental health than their counterparts. In this study, the health science disciplines refer to nursing and allied health studies. Although arts students have more leisure time to engage in physical activities, students from health science disciplines enjoy a better mental health status. The curriculum for the health sciences includes discipline-specific subjects as well as subjects related to the social sciences, such as psychology, sociology, healthy lifestyles, health promotion, communication, and counselling. In the early years of a health science student’s studies, foundation subjects on healthy lifestyles, psychology, and sociology are taught to enable students to acquire knowledge on health-related issues before moving on to subjects on the delivery of patient care. According to the Nursing Council of Hong Kong, the curriculum is designed to develop the personal and professional effectiveness of future nurses [53].

For our population group, resilience and physical activity scores are possible predictors of the positive mental health status. In the previous discussion, correlations were found between the two variables in our study sample. For more than half a century, having the ability to take life as it comes and to master it was considered one of the six contributors to positive mental health [54]. The WHO defines positive mental health as a positive emotion that embraces self-esteem, resilience, and the ability to cope with stressors [4]. The characteristic of resilience also helps to explain the variance in the cumulative GPA of undergraduate college students, in addition to aptitude and achievement [13]. Other variables such as the number of hours that one engages in a part-time job per week and the number of intimate friends that one has cannot be entered into the final model. More research is needed to interpret why a part-time job and intimate friends were non-significant factors in the positive mental health status of the participants.

There are some limitations in this study. The use of a non-random sampling method means that the results of this study cannot be generalized to all college students, particularly students in other parts of the world. The students who took part in this study may not be representative of the entire student population. In addition, this study was cross-sectional in design, which eliminates interpretations of cause and effect regarding the variables of interest. The brevity and self-reported nature of the measurement scales do not capture all of the constructs in their entirety. Future studies should incorporate broader measures of wellness for a comprehensive measure of the mental health status.

## 5. Conclusions

This study showed that college students in different disciplines have different exercise or physical activity patterns. Students in the health sciences have a more positive mental health status than students in non-health science disciplines because of the broad curriculum required for professional training in the health sciences. Lastly, greater resilience, higher physical activity scores, being male, and being a student in a health science discipline are predictors of a better mental health status. Further studies are needed to investigate other issues that may affect the positive mental health status of freshmen, for the purpose of planning health promotion programmes.

## Figures and Tables

**Table 1 ijerph-16-03210-t001:** Students’ demographic characteristics (*n* = 416).

Students’ Demographic Characteristics	No. (%)
Gender	
Male	167 (40.1%)
Female	249 (59.9%)
Study discipline	
Arts	47 (11.3%)
Sciences	78 (18.8%)
Social Sciences	61 (14.7%)
Health Sciences	230 (55.3%)
Part-time job (hours/week)	
Nil	132 (31.7%)
<8 h	102 (24.5%)
8–16 h	115 (27.6%)
17–24 h	52 (12.5%)
>24 h	15 (3.6%)
Number of intimate friends	
0	17 (4.1)
1–2	142 (34.1)
3–5	182 (43.8)
6 or above	75 (18)

**Table 2 ijerph-16-03210-t002:** Scores on the resilience, positive mental health, and physical activities (*n* = 416).

Scores	Minimum	Maximum	Mean (SD)
Brief Resilience Scale (BRS)Score	10	30	18.815 (3.33)
Positive Mental Health Scale (PMH-Scale) Score	10	36	26.248 (5.116)
Godin-Shephard leisure-time physical activity questionnaire (GSLTPAQ) Score	0	200	44.050 (26.723)

**Table 3 ijerph-16-03210-t003:** Correlations between the physical activities, mental health, and resilience.

Scores	BRS Score	GSLTPAQ	PMH-Scale Score
BRS Score	1	0.172 *	0.485 *
GSLTPAQ		1	0.258 *
PMH-Scale Score			1

* *P* < 0.01.

**Table 4 ijerph-16-03210-t004:** Comparisons between students from different disciplines on physical activities, mental health, and resilience.

Comparisons of Scores from Students in Different Disciplines	F	Sig
BRS Score	1.389	0.246
PMH-Scale Score	2.488	0.06
GSLTPAQ	3.838	0.01 *

* *P* < 0.01.

**Table 5 ijerph-16-03210-t005:** Post hoc test for GSLTPAQ among students of different study disciplines.

Study Disciplines		Mean Difference	Sig
Arts	Sciences	15.972	0.006 *
	Social Sciences	13.660	0.04 *
	Health Sciences	11.572	0.033 *

* *P* < 0.05.

**Table 6 ijerph-16-03210-t006:** Multiple Regression Analysis for predictors of Positive Mental Health

Variables	β (95% CI of β)	*P*-Value
BRS total score	0.704 (0.575–0.833)	* < 0.001
GSLTPAQ total score	0.032 (0.016–0.048)	* < 0.001
Male (reference group: Female)	1.035 (0.171–1.900)	* 0.019
Health Science Students (reference group: Non-health sciences)	1.052 (0.175–1.930)	* 0.019
Hours of part-time job/week	0.014 (−0.358–0.386)	0.942
Number of intimate friends	0.374 (−0.174–0.922)	0.181

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
