# Peer review of "Assessing the Mental Health, Physical Activity Levels, and Resilience of Today’s Junior College Students in Self-Financing Institutions"

_ijerph, 2019, doi:10.3390/ijerph16173210_

Round 1

Reviewer 1 Report

The authors evaluated the association between physical activity, mental health and resilience in a sample of college students in Hong Kong, reporting significant associations between mental health and predictors such as resilience, physical actiivty and enrollment in health science disciplines.
The study is of interest and well written, the methodology appears to be sound.

I only have a minor suggestion:
- the authors should report whether they checked if the data met the assumptions necessary to conduct regression analysis (e.g. normality of residuals, and homoscedasticity).

Author Response

Response: Due to large sample size, the normal probability plot was used to examine data normality of the dependent variable. The dependent variable is roughly normally distributed overall. The assumption was fulfilled before data analysis began. The above statements were added into the revised manuscript.

Reviewer 2 Report

There is potential to contribute to understanding of positive mental health in college/university populations.  I am not sure, even if revised, that this paper will provide enough originality and scientific rigour for IJERPH readers.

This is a cross-sectional design and it is therefore difficult to interpret the results.  Many of the associations may be bidirectional or due to other factors that have not been measured.  All of this is unknown.  There isn't a theoretical model e.g. of how positive mental health develops or is maintained, used to make predictions. I believe this is definitely needed to understand why and how positive mental health, physical activity and resilience might be related.  Deci & Ryan's Self-Determination Theory is the most obvious, although it might be difficult with the measures used in the present study.

The focus on positive mental health is overshadowed by the reference to mental health problems such as anxiety, suicidal ideation, depression etc throughout the manuscript.  It needs to be quite clear that the focus is on positive mental health (i.e. that's what was measured). 

Discussion of mental health problems makes the manuscript confusing.  The justification for predictions e.g. between physical activity and mental health are based on research relating to mental health disorders rather than positive mental health.  There is also evidence in other sections where it seems that mismatched studies have been discussed without acknowledging the differences in constructs measured.  e.g. Lines 304-306 compare the present finding of no association between resilience and physical activity with a study examining exercise and acute social stress.   

The objectives described lines 106-110 are not quite the same as the aim identified at the beginning of the Discussion.

There are many unsubstantiated claims within the manuscript.  For example, on lines 363-364 it's claimed that arts students have more time for leisure activities than health students, I couldn't see how this was measured in the study and cannot be assumed.  Line 351-352 refers to girls turning to peers for support and peers not having effective strategies (no citation).  Use of a theoretical model will give the discussion a better focus and reduce the need to make unsubstantiated claims to explain the findings.

The limitations need to acknowledge the brevity and self-report nature of the measures used.

Because this is a cross-sectional study, it is important to be very clear about what associations and predictors.  With the current design/findings it is not possible to make strong claims about predictors.  Could it be the case that physically active males with higher positive health scores are more likely to enrol in health sciences?  There are many reasonable interpretations of what these associations mean and it's possible to speculate on causality if there is a theoretical model driving the hypotheses - otherwise it isn't wise to go beyond reporting associations.  It is also possible that revision of the manuscript will mean presenting/analysing the data in a different way.

I hope these comments are useful.

Author Response

There is potential to contribute to understanding of positive mental health in college/university populations.  I am not sure, even if revised, that this paper will provide enough originality and scientific rigour for IJERPH readers.

This is a cross-sectional design and it is therefore difficult to interpret the results.  Many of the associations may be bidirectional or due to other factors that have not been measured.  All of this is unknown.  There isn't a theoretical model e.g. of how positive mental health develops or is maintained, used to make predictions. I believe this is definitely needed to understand why and how positive mental health, physical activity and resilience might be related.  Deci & Ryan's Self-Determination Theory is the most obvious, although it might be difficult with the measures used in the present study.

Response: Though a theoretical model that links physical activity, resilience and mental health is not available, the literature review shows a relationship between positive mental health and resilience, physical exercise and resilience, and physical exercise and positive mental health. It is the purpose of the present study to sorting out the connections of these constructs using correlation and regression analysis. The results pave the way for future studies for developing a conceptual model that depicts the relationships.

Thank you for introducing the Self-Determination Theory. From the literature review, the above theory explains the relationships between physical activity, motivation and social physique anxiety. The authors decided that the Theory will not be referenced as the theoretical model of the study.

The focus on positive mental health is overshadowed by the reference to mental health problems such as anxiety, suicidal ideation, depression etc throughout the manuscript.  It needs to be quite clear that the focus is on positive mental health (i.e. that's what was measured).  

Response: Thank you for the constructive comments. The definition of positive mental health was defined in line 58. To avoid being shadowed by mental health problems, the references for anxiety have been reduced, while the relationships between physical activity and positive mental health have been included.

Discussion of mental health problems makes the manuscript confusing.  The justification for predictions e.g. between physical activity and mental health are based on research relating to mental health disorders rather than positive mental health.  There is also evidence in other sections where it seems that mismatched studies have been discussed without acknowledging the differences in constructs measured.  e.g. Lines 304-306 compare the present finding of no association between resilience and physical activity with a study examining exercise and acute social stress.  

Response:  Please refer to the above responses. Part of the discussions on “The correlations between physical activities, mental health, and resilience” were rewritten. Several references were changed to avoid the mismatched studies. The reference for Line 304-306 was removed. Thank you.

The objectives described lines 106-110 are not quite the same as the aim identified at the beginning of the Discussion.

Response: The aim identified at the beginning of the Discussion was revised.

There are many unsubstantiated claims within the manuscript.  For example, on lines 363-364 it's claimed that arts students have more time for leisure activities than health students, I couldn't see how this was measured in the study and cannot be assumed.  Line 351-352 refers to girls turning to peers for support and peers not having effective strategies (no citation).  Use of a theoretical model will give the discussion a better focus and reduce the need to make unsubstantiated claims to explain the findings.

Response: We did not measure the leisure time in this study. However, detailed explanations were provided such as arts students do not need to complete as many credits and not required to complete a practicum as a condition for graduation. This could partly explain why arts students are having a less busy school life resulting in more leisure time.

For girls turning to peers for support, the in-text citation was moved to explicitly referring to the statement, instead of referring to the entire sentence.

The limitations need to acknowledge the brevity and self-report nature of the measures used.

Response: The above suggestion was included.

Because this is a cross-sectional study, it is important to be very clear about what associations and predictors.  With the current design/findings it is not possible to make strong claims about predictors.  Could it be the case that physically active males with higher positive health scores are more likely to enrol in health sciences?  There are many reasonable interpretations of what these associations mean and it's possible to speculate on causality if there is a theoretical model driving the hypotheses - otherwise it isn't wise to go beyond reporting associations.  It is also possible that revision of the manuscript will mean presenting/analysing the data in a different way.

Response: Though a theoretical model is not available to conclude about gender and higher positive health scores, there are literatures to support girls are having lower positive health status than boys. The above evidence has been included in the revised manuscript. The authors agreed that it is not possible to make strong claims about the predictors, the statement has been included to guide the readers in this regard. We also stated that those are suggested predictors due to the study design.

Reviewer 3 Report

The manuscript entitled “Assessing the mental health, physical activity levels, and resilience of today’s junior college students in self-financing institutions” deals with the causes of the disorders in mental health among students of Hong Kong. By using a written questionnaire, a statistical analysis is carried out with the answers finding that, art students are more prone to have physical activity, and that this factor, along with resilience, is considered as a reliable factor for mental health.

The assessment is not clear, however, at the methodology and the description of the results. There is not a clear correlation between physical activity and the mental status, at least at first sight. Please be clearer and more detailed in the description of the analysis. Otherwise, the document is not worth to be published. For instance, in the answers sheet there is no an answer regarding physical activity, how come the results show a relationship between physical activity of the students and a higher mental health?

Specific comments:

Line 58-59: The concept of mental health was already described in line 34.

Line 121: I don’t know whether “greater” is the proper word.

Line 27: The equation doesn’t have numbering. Also, it is not equaled to anything.

Line 173-177: What’s the basis of switching the activities? Why cross-country is considered equivalent to jumping rope? There is an analysis of the metabolic rate?

Line 185: “construct”?

Line 196-197: The statement was already mentioned in line 136.

Line 200-201: 5% of the questionnaires were not answered, why the response rate is 92.4%?

Line 212: “more were female (59.9%) than male”. Rewrite the sentence.

Line 254: “gender” rather than “sex”.

Line 373-390: In my opinion, this part should be at the Conclusion section.

Author Response

The manuscript entitled “Assessing the mental health, physical activity levels, and resilience of today’s junior college students in self-financing institutions” deals with the causes of the disorders in mental health among students of Hong Kong. By using a written questionnaire, a statistical analysis is carried out with the answers finding that, art students are more prone to have physical activity, and that this factor, along with resilience, is considered as a reliable factor for mental health.

The assessment is not clear, however, at the methodology and the description of the results. There is not a clear correlation between physical activity and the mental status, at least at first sight. Please be clearer and more detailed in the description of the analysis. Otherwise, the document is not worth to be published. For instance, in the answers sheet there is no an answer regarding physical activity, how come the results show a relationship between physical activity of the students and a higher mental health?

Response: The heading of 3.2 has been revised to “Resilience, mental health and physical activities of student” to align with the results in Table 2.

Sorry there were mistakes in the discussion of correlations. “A significant positive weak correlation between resilience and physical activities (r=0.26)” was wrong. The statement has been changed to “A significant positive weak correlation between mental health and physical activities (r=0.26)”.

The statement “There was no correlation found between physical activities and resilience” was added in the Result section for clarity.

The Discussion section for correlations between physical activities, mental health and resilience are correct after thorough checking.

The answer regarding physical activity was displayed in Table 2.

Specific comments:

Line 58-59: The concept of mental health was already described in line 34.

Response: The repeated information has been removed.

Line 121: I don’t know whether “greater” is the proper word.

Response: “greater” has been changed to “wide range”.

Line 27: The equation doesn’t have numbering. Also, it is not equaled to anything.

Response: The numbering has been added into the equation in the revised manuscript. It is equaled to the sample size. “Sample size” has been added before the equation.

Line 173-177: What’s the basis of switching the activities? Why cross-country is considered equivalent to jumping rope? There is an analysis of the metabolic rate?

Response: A few of the activities are considered not appropriate for the Hong Kong people due to cultural differences and geographical constraint.  According to the activity information from Department of Health, Hong Kong, jumping rope is classified as strenuous exercise. As Hong Kong students may not have ideas on “cross-country skiing”, it was replaced by jumping rope as most students are familiar with.

There is no analysis of the metabolic rate. However, the Department of Health made references from the “Pacific Physical Activity Guidelines for Adults, World Health Organization Regional Office for the Western Pacific Region, 2009” to determine the kind exercises that are classified as vigorous, moderate and low intensity. The above information can be found in the website of Department of Health, Hong Kong, which is included in the reference list.

Line 185: “construct”?

Response: “construct” has been replaced by “dimension”.

Line 196-197: The statement was already mentioned in line 136.

Response: The statement removed.

Line 200-201: 5% of the questionnaires were not answered, why the response rate is 92.4%?

Response:
The statement was rewritten as “The questionnaire was discarded if more than 5% of the questions were not answered” to avoid confusion.

Line 212: “more were female (59.9%) than male”. Rewrite the sentence.

The sentence was rewritten as “There were more female students (59.9) than male”.

Line 254: “gender” rather than “sex”.

Response: Changed.

Line 373-390: In my opinion, this part should be at the Conclusion section.

Response: As the second last paragraph is the continuation on the discussions of regression analysis, while the last paragraph discusses the limitation of study, the authors decided to keep these ideas in the discussion section.

Round 2

Reviewer 2 Report

The revised version is a major improvement.  My only concern is that there continues to be a claim about arts students having more leisure time based on their course credits compared to students in other degrees.  It's not a claim that can be supported by the data collected.  We don't know, for example, if arts students have more onerous assessment or take on other activities on campus.  It seems to be based more on a stereotype of arts degrees rather than evidence and the comment should be removed.

Reviewer 3 Report

Most of the comments are already addressed. Nonetheless, the comment regarding Equation 1 and 2 remains: I don’t see any numbering nor any equalization to the “sample size” term.